# The Racial and Ethnic Identity Development Process for Adult Colombian Adoptees

Veronica Cloonan [1,*], Tammy Hatfield [2], Susan Branco [3] and LaShauna Dean [2]

1  Independent Researcher, Gaithersburg, MD 20878, USA
2  Counselor Education, University of the Cumberlands, Williamsburg, KY 40769, USA
3  Counselor Education, St. Bonaventure University, St. Bonaventure, NY 14778, USA; sbranco@sbu.edu
*  Correspondence: doctorcloonan@gmail.com

**Abstract:** This research aimed to understand the process adult Colombian adoptees raised in the United States of America go through to define themselves in the context of race and ethnicity. The research followed a qualitative narrative methodology, in which six participants were interviewed twice regarding their experiences with transracial and transnational adoption and their ethnic and racial identity process. The results suggest that identity is a dynamic process. Our research also confirms Colombian's history of unethical adoptions and its influence on the complexity of identity and loss of adult Colombian adoptees. Throughout the article, the researchers use the term biological family referring to Colombian birth families. However, we acknowledge that other terms (i.e., first, natural, original, etc.) are also used in the adoptee community.

**Keywords:** Colombian adoptees; transracial adoption; racial identity; ethnic identity; narrative methodology

## 1. Introduction

Adoption in the United States of America has a long history. Starting in colonial times (1750–1800), adopted children were seen as a source of free labor and service for adoptive families (Javier 2007). The domestic adoption practices transitioned to international adoption after World War II. The war left orphan children, and Americans showed interest in adopting them. This practice became normalized even more with the Indian adoption project in the 1960s, where Native American children were placed with White families due to the painful colonization history (Adoption Network 2022). International and transracial adoption in the United States of America had an evident pattern; minoritized races were being adopted by mainly White American families with minimal to no cross-country regulations (Flores-Koulish and Branco Alvarado 2015; Branco-Alvarado et al. 2014; Javier 2007; Kim et al. 2010; Samuels 2009; Cloonan et al. 2023). Marketing and social media often portray domestic and international adoption from the savior and happy-ending approach. The transracial adoptee is often placed in a position where they are expected to be faithful to their adoptive family above any links they may try to reclaim with biological kin (Sohn 2022). International adoption can create identity dissonance when the adoptee faces the challenge of making changes in multiple cultural affiliations (Baden et al. 2012; Branco et al. 2022). Identity is a core issue adoptees face and lies at the heart of the lived experiences of adoption (Baden and Steward 2007). Scholars have viewed choosing a racial label that does not match your appearance as maladaptive (Baden and Steward 2007). Advocacy efforts promote transracial adoptees to claim more than one racial and ethnic identity given the complexities of their lived realities. Colombian and transracial adoptees in the United States of America encounter unique challenges when raised by White American parents who do not share the adoptee's culture, language, or heritage. Their biological background is from a different country, from a population comprised of different races and cultural compositions (Brookover 2023; Cloonan et al. 2023). The adoptive parent should support

the child to acquire knowledge and pride in their race, ethnicity, and culture to support the child's identity development associated with their birth family and background (Waterman et al. 2018).

Colombia has practiced international adoption for over 50 years. Some Colombian adoptions were ethical and legal, following intercountry regulations and getting proper consent without coercion (Smolin 2021). Many other Colombian adoption practices were illicit, especially those between the 1960s and 1990s (Branco 2021; Carreazo 2016; Kawan-Hemler 2022). International adoption practices can result in child trafficking as it involves transferring them from poor, less-developed nations to rich, first-world countries. Often, international adoption involves a high monetary price, and its purpose is to meet the demands of those rich nations and families for children (Cheney 2021; San Román 2021; Smolin 2004, 2010, 2021). International adoption has had long documented facts of child trafficking, leading the US State and the Federal Bureau of Investigations (FBI) to explore adoption irregularities around 2017 (Cheney 2014, 2021; U.S. Department of Justice 2020). International adoptions have moved from one sending country to another as irregularities are noted and regulations are implemented to protect children (Cheney 2021). The measures to protect against international child trafficking started by the Hague Convention in 1993 have been insufficient (San Román 2021). Smolin (2021) advocates stopping international adoptions completely until they can be carried out correctly and meet international requirements, including remedies for those past international adoptions conducted fraudulently.

At present Colombian adoptions are more heavily regulated than 50 years ago (Villa Guardiola et al. 2022), especially after the Colombian government signed the agreement for the Hague Convention in 2012 (Alayon 2014; The Hague Convention n.d.). However, information on current adoption practices is challenging to find. The Instituto Colombiano de Bienestar Familiar (ICBF) is the child welfare agency in Colombia. The ICBF reportedly currently prioritizes in-country adoptions while international adoptions are for older children who cannot be placed with extended family members and who are at risk of aging out of the system (Bautista-Lopez 2016). Of countries that belong to the Hague Convention, Colombia is still one of the top countries sending children for adoption to the United States, along with India and Bulgaria (The United States (U.S.) Department of State 2022).

We share the research results on the narratives the adult Colombian adoptee participants built while trying to make sense of their international and transracial adoption. Colombian adoptees are responsible for defining and integrating their racial and ethnic identity to reflect their unique experiences of transracial adoption. In addition, unlawful practices of Colombian adoptions have been well documented (Branco and Cloonan 2022; Branco 2021). When illicit adoption occurs, falsification of documents and inaccurate information is often found in the adoption history. Transracial adoptees face challenges regarding their identity development due to the fabricated or absent information on their birth family (Baden et al. 2012; Park-Taylor and Wing 2019). Throughout the article, the researchers use the term biological family referring to Colombian birth families. However, we acknowledge that other terms (i.e., first, natural, original, etc.) are also used in the adoptee community.

In this research, ethnic and racial identity is understood as two undivided but separate concepts, especially when talking about Latinx or Colombian adoptees. In the United States, there is a prevalence of "racialized ethnicities" [Colombians in New York] and ethnicized races" [Blacks in the United States] (Gallegos and Ferdman 2012). Racial identity is then understood as the personal sense of belonging to a racial group based on physical and psychological components (Cloonan 2022; Helms 2008). Ethnic identity is understood as how the individuals self-label and their sense of belonging to an ethnic group with a common national or cultural tradition (Cloonan 2022; Phinney 1990; Fisher and Lerner 2005; Waterman et al. 2018). For the current study, the lead researcher identifies as a Colombian native and immigrant to the United States. They are a practicing Licensed Clinical Professional Counselor with significant experience working with transracial and

transnational, including Colombian adoptees. The other research team members include a full-time professor and two assistant professors in counselor education. The third author is also a transracial and transnational Colombian adoptee.

## 2. Materials and Methods

The researchers aimed to collect the narratives of Colombian adult adoptees raised in the United States to answer the research question: How do adult Colombian adoptees construct their racial and ethnic identity? The participants completed two semi-structured interviews with the lead researcher. Questions posed related to their racial and ethnic identity process, cultural socialization, and the impact of international adoption in their racial and ethnic identity process.

### 2.1. Research Design

The qualitative study utilized narrative inquiry because of the unique and specific phenomena of the experiences of how Colombian adult adoptees assign meaning to their experiences through the stories they told during the research process (Bloomberg and Volpe 2012). A narrative methodology is also suitable for research on topics of identity and identity development because it allows participants to share their life narratives and meaning-making process about themselves (Lieblich et al. 1998; Butina 2015; Kim 2016). The chronological story was the primary foci of the study, and a narrative methodology qualitative approach allowed the stories and histories of the participants to become the raw data that ultimately answered this study's research question and sub-questions.

### 2.2. Participants

After obtaining Institutional Review Board approval, the researcher recruited adult Colombian adoptee participants adopted between the ages of birth and 12 years old by United States parents. The participants were recruited through a Facebook group for Colombian adoptees with 2.7 K members. The sampling strategy was a purposeful, homogeneous, and convenience sampling process. In total six (n = 6) adult Colombian adoptees: three identified as women and three identified as men. Although the sample size was small, the data collection was extensive due to the rich descriptions and information gathered during the in-depth interviews required by the narrative approach. Please refer to Table 1 for detailed information about the participants.

**Table 1.** The Participants.

| Participant [1] | Place of Birth | Age of Adoption | Adoptive Parents' Race | Place Where They Grew Up | Age |
|---|---|---|---|---|---|
| Gonzalo | Bogota | 5 months | White | Midwest, USA | 33 |
| Andres | Cali | 7 months | White | West Coast, USA | 37 |
| Valentino | Bogota | 5.5 months | White | East Coast, USA | 34 |
| Juliana | Bogota | 4 months | White | East Coast, USA | 36 |
| Camila | Bogota | 4 months | White | East Coast, USA | 28 |
| Carmen | La Mesa | 6 years | White | Midwest, USA | 36 |

[1] Participants chose their Pseudonyms.

### 2.3. Data Collection Methods

The questions developed for the semi-structured interview emerged from the researcher's clinical experience working with international adoptive families, the literature review, and input from different content experts to include Colombian adoptees in the United States. The data were collected using semi-structured interviews in the English language and recorded via a secure Teams platform. Ongoing collaboration occurred throughout the study via member checking between the participants and the researcher.

Additionally, member checking occurred throughout the study to ensure the participants' narratives were accurately reflected. Member checking allowed for further questions about emerging themes and the opportunity for a third interview if needed.

### 2.4. Data Analysis Process

The process of data analysis was both deductive and inductive. The initial categories of the conceptual framework were deductively obtained from previous research and theoretical models (Bloomberg and Volpe 2012). During data analysis, the researcher used sequencing-chronological data as the primary method. Chronological sequencing was used for coding the story told by the participants during the first coding cycle. The data analysis process was completed in five steps, as explained in Figure 1. The participants collaborated with the researcher to increase the report's trustworthiness and accuracy.

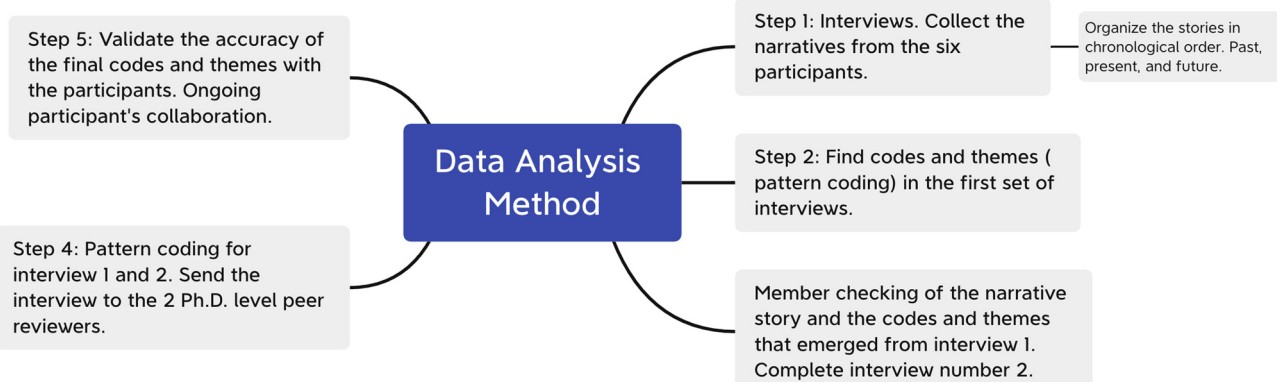

**Figure 1.** Data Analysis Process.

The first step was to collect data through oral interviews. The researcher organized the data chronologically, built in place and setting, described the story, and completed step two, identifying the first draft of codes and themes. The third step refers to retelling the individual's story. From this approach, the researcher contacted the participants again and told them their stories as the researcher had understood and compiled them. The researcher presented their written chronological stories, and the participants and researcher made the necessary edits. During this step, the second interview was also completed. After the participants' approval, the researcher went to step four, which refers to finalizing the pattern coding of interviews one and two. Each coded interview was sent to the participants to verify the codes' accuracy. Two interviews were also sent to the peer debriefers to compare the codes from the researcher and colleagues. Finally, this leads to step five, validating the report's accuracy and final themes after the codes were placed in themes and categories. Once again, this was conducted in collaboration with the participants. At that time, a final story was presented in chronological sequencing with information about codes that became themes and subthemes. The chronological story was organized in context, place, and setting.

### 2.5. Measures for Ensuring Trustworthiness

The researchers addressed trustworthiness by establishing validation strategies for credibility, dependability, confirmability, and transferability. Throughout the research, we utilized the member-checking technique, reflexive journaling, and peer examination of the data or trustworthiness measures.

### 3. Colombian Adoptees and Their Racial and Ethnic Identity Process

International adoption is a multi-part process involving the adoptive parent, the biological parent, the child, and the adoption agency. Post-adoption resources can be limited in the United States, and children and families face the difficult task of assimilation



into the new and unique family dynamics with little support (Baden et al. 2013; Dekker et al. 2017; Lamanna et al. 2018) Scholars have advocated for more comprehensive training and transparency from adoption agencies to foster social justice within adoption practices (Branco 2021; Mounts and Bradley 2020). Many adoptees experience both loss and gains through the act of adoption (Roszia and Maxon 2019). They experience unique issues, including establishing a relationship with the new family, forming a coherent identity, and grieving the loss of a biological family, culture, and nationality. The research findings highlight how participants navigated racial and ethnic identity formation as Colombian adoptees. Table 2 offers an overview of the four main themes gleaned from participant interviews. Participant quotes highlighting each theme and related subthemes follow.

**Table 2.** Themes and Subthemes for Racial and Ethnic Identity.

| Themes | Definition | Participants [1] Who Endorsed It |
|---|---|---|
| Theme 1: Putting the puzzle together. "It is a Dynamic Process". | When racial and ethnic identity changes throughout the years. | Camila, Juliana, Andres, and Valentino. |
| Theme 2: Brown on the outside, White on the inside. | When participants grew up in a White family, but they do not look like they are White. | Camila, Juliana, Valentino, Gonzalo, and Carmen. |
| Theme 3: A chameleon with imposter Syndrome. | When the participant can adapt to the Colombian and White environment without feeling like they fully fit in either of these spaces. | Andres, Gonzalo, Valentino, and Juliana. |
| Theme 4: Adoption as a loss. | When adoption is experienced as a loss or as a traumatic experience. | Camila, Valentino, Carmen, Andres, Juliana, Gonzalo. |

[1] Participants chose their Pseudonyms.

*3.1. Putting the Puzzle Together: "It Is a Dynamic Process"*

Participants' responses exemplified how their racial and ethnic identity process was dynamic. Participants reported the process was influenced by context, social interactions, and messages regarding their race and appearance. Camila stated that talking about race and ethnicity growing up was very challenging for her. Camila also reflected on how much her racial and ethnic identity process changed throughout the years. She said:

> I think when I was a little kid, I really couldn't talk about anything and was very shut off about being Colombian, obviously, I knew that, but I didn't really wanna talk about race . . . I think it's changed a lot since I was a kid versus now.

Juliana reported that for her, the racial and ethnic identity process changed and moved forward in a way that she has more clarity about now. She reported: "I think there's just been more clarity".

Andres reported that growing up, he was not aware of Black people speaking Spanish, he said: "I was completely ignorant to that reality. And now happily claiming it". When it came to what influenced his process of racial and ethnic identity, he reported: "I think at the very beginning of when I first went to Colombia and then coming back here, and then hanging out with a lot of Mexican-American immigrants, Peruvian, mostly Peruvian and Mexican. The word 'Mestizo' [a person of European and indigenous ancestry] kept coming up". When he reflected on how his racial and ethnic identity has changed, he said: "Before like denying any kind of Blackness, but not for any kind of malicious intent, just because I didn't know, I didn't know any better". For Andres, the racial and ethnic identity process allowed him to claim his African descent.

Valentino reflected on how his racial and ethnic identity process remains dynamic and fluid for him. He stated: "The process of going through it has been fluid, if anything. So, it

seems that I cannot prescribe still to any one thing, and it's still, like you said, a mixture of an amalgamation of all these different ideas and identities". He then added, "It has certainly changed throughout the years". He also reflected on how "That [racial and ethnic identity] has changed as of late". Since he conducted a DNA test, he found out that he was of 50% Indigenous ancestry.

### 3.2. Brown on the Outside, White on the Inside

Participants discussed their experience of life as people of color who spend most of their time in White spaces. All of the participants identified as a person of color, all were also raised in a White family, attended primarily White schools, and, as children and adolescents, socialized mainly with White people. Valentino says: " . . . Because as much as I might feel something inside, there is also the reality of how I am, how I look to the outside world, so I cannot escape that".

In this theme, three sub-themes emerged. The first one was their clarity about not being White, the second one referred to racist experiences, and the third one to the racial isolation they grew up with.

### 3.2.1. "I Am Not White"

Four participants frequently stated, "I am not White". Participants reported this sentiment was especially prevalent when they had to answer questions regarding their race and ethnicity in the US Census or demographic questionnaires. Participants indicated they did not feel like any of the options in demographic questionnaires resonated with the complex answer of race and ethnicity.

### 3.2.2. Racism

This subtheme arose when the participants shared their experiences with racism. Many participants described how they frequently received direct or implicit aggression from the White and Hispanic communities. Participants reported they were often "not White enough" and "not Hispanic enough". Many were criticized by members of both White and Hispanic communities for not speaking Spanish or being knowledgeable about Colombian culture and cuisine traditions.

When talking about racism and discrimination, Carmen reports:

*For the longest time, I really didn't identify myself as a person of color because I felt like I had grown up into a such a Caucasian household, but as I grew older into adulthood and I started talking to other people of color, I realized that I guess I was technically a person of color, I wasn't considered to be someone of African-American descent or any of that, but I realized I was a minority and I maybe didn't have it as rough as other people of color might have, but I still was facing discrimination.*

Carmen also reports she felt "not Colombian enough":

*Well, like with ethnicity, I felt I had it because I felt like when I was around a lot of Colombians, that I wasn't really Colombian enough. I noticed that one day we were invited, my husband and I were invited to a dinner from a friend of mine up in Fargo, and she was from, I believe, Bogota, and she knew a bunch of people here that were also Colombian. And so she invited me over there and I was looking forward to it, but then when I got there, I just felt this immediate disconnect because everyone there assumed that I spoke Spanish and when they found out that I didn't speak Spanish, they didn't wanna speak to me anymore.*

Andres reports he also experienced discrimination when it came to race and ethnicity, he stated:

*And then also Colombian immigrants here in the Seattle area who [chuckle] who would claim that because I didn't speak Spanish and that because I wasn't raised in Colombia, that I had no right trying to call myself Colombian, trying to wear "la camiseta" [Colombian's soccer team shirt] during games. Right? Like, "Who do you think you*

*are, you a nerd? You've never lived in Columbia, you weren't raised there, you don't speak Spanish". "No tienes derecho". It's what they would always say. And it wasn't happening all the time, only when I would go to like a de julio celebration here or like a . . . Yeah, a soccer game, and I would come with my jersey on or something and . . . Or with a flag or something [chuckle] And sometimes people would say, "Oh yeah". And they started speaking to me in Spanish, and it's like, "Oh no, sorry. Lo siento. I don't speak Spanish". Yeah, yeah. And the response was, many times, "Oh, then, what are you doing here? You're not really Colombian. You're not Colombian enough".*

### 3.2.3. Racial Isolation

Participants reported that they experienced significant racial isolation with little, if any, racial mirrors (people who look like themselves). Camila reported, "I really don't look anything like my mom and my dad that adopted me". The participants reflected that being adopted was something obvious that people could notice. Andres reported, "There were no opportunities for what I've heard referred to as racial mirrors. And so, I didn't have anyone, no teachers that looked like me, no . . . There were some people that looked like me, but they were all on TV".

### 3.3. A Chamaleon with Imposter Syndrome

In this theme, participants reported they can "blend in" in multiple spaces. "It depends on the crowd," Andres said. Participants indicated that they can understand the White-American perspective, they can understand the Colombian perspective, and a person of color perspective. They can also understand their ethnicity, and how they were raised. Some participants' families were Puerto Rican, Italian American, Eastern European, etc. Participants described having the experience of being a "chameleon" and feeling, at times, like an imposter. They further clarified that when they try to claim an identity (the Colombian identity, for example), it feels like it is an undeserved or illegitimate identity.

Valentino, who recently found out he is 50% Indigenous, reports:

*Now I'm kind of fluctuating between all three, but lately, I've been using the Indigenous one, the Native-American sort of one. So, I almost get to pick and choose, daily, which one I want to be, which is in its own way nice and at the same time it's, again, no less confusing. So, it went back and forth. It did, but it always changed, it kinda would change to suit the situation. After 10 years old, it just started going back and forth, like a tennis match.*

Camila explained that she can "kind of feel like a chameleon sometimes, 'because I can understand where White people are coming from about 'because I grew up around that culture, but then at the same time, I'm like, 'I'm not cool with it.'" She also reported that

*In terms of race, I would say I have Indigenous ancestry and Spanish, so White-European, so a mix of those two. And I typically have a lot of issues with the way the US census puts race down. I always have a lot of issues with that because if I say I'm Native American, they're gonna think I'm from one of the tribal nations in the United States, but we all know that in the Americas, there's multiple Indigenous groups, and then if I put . . . I don't identify as Caucasian, so yeah, so I guess it's like how I'd say race.*

### 3.4. Adoption as a Loss

Participants described the adoption experience as a terrible loss. Camila said, when referring to the ethnic and racial process, "If I was born there and left as a baby, I don't have anything to work off of". Valentino reported many people expect him to be thankful for adoption, but overall, it is the feeling of loss that he experiences the most. He said, "And they also tell me that they are happy how things progressed for me. I still can't get away though from the feeling that I was still ripped away from my family".

Carmen reported that adoptees go through a lot of losses. They go through the loss of a family, a name, a country, and a culture, and the losses keep piling up throughout the years. As she put it when describing her decision not to change her name for her marriage,

"I didn't want to give up my name because I felt like I had already had this name for such a long time and then to have me give it up for his name would be asking a lot, because I had already given up my previous name when I came here".

Many adoptees experience a change of their names and last names when they come to the United States. This also impacts identity development since they often lose their Latino last names as one of those connections to Colombian roots. Three subthemes emerged: Unethical adoptions, ethical adoptions, and internalized symptoms.

### 3.4.1. Unethical Adoptions

This sub-theme captures those participants who reported an unethical adoption and/or noted many discrepancies and even opposite information in the adoption documents than what was reported from their biological family. Andres reported:

> Well, there's two versions, there's the version that's in the adoption documents and then there's the version that she told me, and they are opposite. The version that she told me was that that's not true, that she never gave me up for adoption. She wanted to send me to live with her sisters and her mother in Tumaco like she did with my brother, but then she said that social workers took me and then promised her that she could go back and get me after a few days after she healed 'because I was a C-section baby. And she went back and then I was gone by that point. My bio mom said, well, there's nothing I could do. And I didn't have money to pay a lawyer to fight the case and so I just hoped for the best.

When talking about international adoption, Valentino reflects on how harmful this process can be for the biological family and the adoptive child. He adds:

> Please let's not do that anymore, especially knowing what actually happened to my original mother, where she was forced to give me up or give up all her kids, I can't see it as anything that is remotely a good thing, at least in that standard sense. The fact that my mother was forced to give me up for adoption because her first husband disappeared in the military, and she was receiving widowed benefits. She had three kids with him, she met someone else, he got her pregnant, and then when he found out, he left. She was told by the government and the military that, "If you don't give up this new kid who's out of wedlock, you'll lose your widow benefits and you'll have to also sacrifice your other three kids, all of them will go into child services". She was also promised apparently, "Okay, so we won't put him up for adoption entirely, he'll go with a sponsor family and after a certain period of time, you can get him back". She was also promised that. She changed her mind after I was . . . Immediately after I was taken out of her arms that first night, she changed her mind, she wanted to actually get me back, she would go back to the orphanage every single day, but they said, "No, you signed over your parental rights, he's no longer yours". Everyone in the organization seems okay at first, once we got back into United States, once they had me, they kept on wanting my adoptive parents to be a part of the organization, from what I was told, almost as a mascot, they wanted people to . . . They wanted them to say what a great story it was and how the process went so smoothly so they can get other people to continuously just try and adopt. It did take them a long time looking at my records to figure out what happened because it seemingly was written, handwritten, my number . . . It certainly was forged, so that happened with me, too. Yeah, so I can definitely see where the corruption is, or was at least between those couple of decades rampant in the country.

Carmen narrated that it was still very confusing for her to understand what happened, why she was adopted, and the whole and accurate version of her story. Carmen stated:

> And from what we were told by our siblings was that my older brother was in charge of watching me and someone had a problem with my mother and called the welfare and they took us. And when they took us, because my mom was uneducated and didn't have any resources or any help, she tried to take us back and they said that they took her rights away and they put us in the system, and we were adopted out. Catholic orphanages were falsifying their documents stating that their children were also abandoned, and they were

*giving them false cedula numbers, they were putting false names on the documents of who their so-called parents were, and so that's why they were having a hard time finding their parents. There was a lot of confusion in it. My parents said in the papers they read us that my brother and I were living on the streets for about a year and then we got picked up and put into foster care. And we learned recently that a lot of the information was just stuff that was put in there because they needed to put information in. And I learned from a lot of adoptees that the term abandonment was used in many documents.*

### 3.4.2. Ethical Adoptions

Conversely, the sub-theme of ethical adoption also appeared in the narratives. For example, Camila reported: "Mine was actually the very, I guess, ethical adoption. Mine was actually really . . . My mom made the choice to do it. She put herself and found a . . . She needed help, her mental health was really bad and was pretty much bad up until, yeah, recently. She's always just struggled with her mental health". When consulting with someone for her to try to find her family, they reported to Camila, "Your paperwork seems legit. It doesn't seem like a corrupted . . . " The consultant reviewing Camila's adoption paperwork reported "Based on what I've seen of other adoptees, this doesn't look corrupt". Similarly, Juliana and Gonzalo highlighted how both of their adoptions were ethical and legal. They also recognized how this was not the norm and how many adoptions were not conducted in this matter through years of Colombian adoption history.

### 3.4.3. Internalized Symptoms

The subtheme describes how participants reported a series of internalized symptoms directly connected to the loss in adoption. Some of the participants reported anger, others depression, and others described strong feelings they experienced while growing up and making sense of their own life stories.

Camila revealed: "I was very angry as a kid, like just angry and couldn't talk about things, unless it was with another adoptee". Valentino Stated:

*Only this past year have I been able to come to terms with why I have always felt . . . I've had depression since I was 10, literally, that's not a hyperbolic. I've had depression that long, where I've always felt unloved, unlovable, and only now, recently in the past year, since my birth family came back into the picture, have I been able to, as a phrase goes, come out of the fog and understand why maybe I really genuinely felt like that, and it really just might be adoption, it might just be being adopted.*

Carmen also reported how the people she worked with in the past were not very understanding of adoption as a loss and the emotional and psychological implications it brings to the adoptees. She adds:

*But the therapist that I saw when I was younger, it was more to deal with my depression. It wasn't until I was older that I realized, this is why I need to see a therapist, because I have identity issues and I need to find out, what is the root to all of this pain that I'm having and this depression and why am I always being put into a depression? I don't know if it made me an emotional eater. I was having a lot of anger and meltdowns and I didn't know why.*

## 4. Discussion

A significant finding of this research suggests that the racial and ethnic identity development of the adult Colombian adoptees participants in this study is a dynamic and ongoing process. Another finding revealed how Colombian adoptees experienced racial isolation and lack of exposure to other racial mirrors. Such scenarios left the participants with the task of integrating their multiple layers of racial and ethnic identity without the resources to do so. For most participants, this resulted in lifelong racial and ethnic identity construction as they continued to explore multiple aspects of their upbringing. Adult Colombian adoptees also reported experiencing racism from the White and Colombian/Hispanic com-

munities. They reportedly felt they were not White nor Colombian enough and endured derogatory comments regarding their race and ethnic "authenticity".

Some participants described feeling like they are putting a puzzle together where all the pieces may not fit with each other as they navigate their racial and ethnic identities. For example, while this study's transracial and international adoptee participants are immigrants, they do not explicitly identify as immigrants. The participants are Colombians (many with dual citizenship), yet they do not fully identify as Colombians or may have feelings of imposter syndrome when trying to claim their Colombian identity. Four out of six participants are still developing their racial and ethnic identities. Some are waiting to go to Colombia to meet their biological families and "get some closure", some are waiting for DNA test results, etc. It is like putting the puzzle together of where they were born, their DNA composition, cultural and ethnic knowledge and customs, and where they want to belong. Some may not know much about Colombia, but they want to know more about their country of origin. While some participants indicated they wanted to learn more about their country of origin, others indicated they knew enough about Colombia but wanted to embrace the US-American culture more. The racial and ethnic identity-development process is unique for each of them and, for most participants, has felt like a puzzle they are trying to fit together.

Our study also reinforced the reality of illicit adoption practices in Colombian international adoption to the United States (Branco and Cloonan 2022; Branco 2021; Carreazo 2016). Half of the participants described unethical circumstances and practices impacting their adoptions which they discovered as adults. Illicit adoption practices impact Colombian adoptees as they alter adoption narratives and sometimes impede their access to biological family information (Branco 2021). The results presented in the current research highlight how ethical or unethical adoption impacts the racial and ethnic identity of adult Colombian adoptees in the United States. Participants reflected on feelings of dissonance and confusion at finding out their adoption stories and documentation did not match or were not found in the system. This confirmed Baden et al.'s (2012) findings regarding transracial adoptees facing additional challenges due to false adoption stories and documentation. Branco's (2021) case study on Colombian adoptees who discovered their adoption documentation was falsified also corroborates our findings on the impact of corrupt adoptions on identity.

Challenges in adoptive families are well documented (Mounts and Bradley 2020). However, there is not much research on parents' quality of training before and after adopting a child. Even though the challenges of these families continue for years after the adoption, services and research in these areas are limited. Future studies should explore how racial and ethnic identity for transracial adoptees shifts when adoptive parents consciously try to address topics of racism and microaggressions while fostering pride and connection with the adoptee's biological roots.

The present research followed different ethical efforts to ensure the accuracy of the data. The researchers were engaged in reflexive journaling and seeking constant participant feedback to ensure data accuracy. Two Ph.D. level-peer debriefers were also consulted during the data analysis to ensure the pattern coding was similar to the results of this research. No discrepancies were found in this data analysis comparison.

## 5. Conclusions

Our findings corroborate other research describing how transnational adoption can impact identity development for the adoptee (Baden et al. 2013; Branco 2021; Branco and Cloonan 2022; Brodzinsky et al. 2022; Mahmood and Visser 2015; Lamanna et al. 2018; Roszia and Maxon 2019). Previous research on identity and adoption determined that when adoptees choose labels that do not coincide with their phenotypes, it is seen as an unhealthy racial identity (Baden and Steward 2007). In this research, the researchers found that all the participants identified themselves as Black, Indigenous, or a person of color (BIPOC). The participants mentioned how it is difficult to choose a racial label or box in the

US census form since many of them are Mestizos or a mix of races that are not encompassed in the responses to demographic questionnaires of the United States.

In terms of their racial and ethnic identity, participants revealed how experiences of racism impacted their ethnic identity conflict and development. According to the participants' narratives, messages about how they look and their Latinx background contributed to them developing the "chameleon" approach, where the participants would sometimes endorse the White or Latinx identity, depending on the context and the people around them.

The current research confirms the unfortunate outcomes of transracial adoptions that often lead to documentation discrepancies, impacting the child's identity (Baden et al. 2012; Branco 2021; Branco and Cloonan 2022). Adoption rights activists have been somewhat successful at increasing US sanctions and judicial reforms (Cheney 2021). Colombian international adoption has decreased after several cases of illicit adoptions were documented (Branco 2021; Branco and Cloonan 2022; Carreazo 2016; Instituto Colombiano de Bienestar Familiar (ICBF) 2022). Intercountry adoptions have also decreased in Uganda, Spain, China, Russia, and Guatemala (Cheney 2021; San Román 2021). However, more advocacy and documented cases of illicit adoption practices across countries are needed to stop new unlawful international adoptions and offer remedies to previous illegal international adoptions (Smolin 2021).

Lastly, our findings confirm that adoptive families need to understand the importance of fostering pride and belonging to the adoptee's birth culture (Waterman et al. 2018). Parents who adopt a child from a different race, culture, and/or ethnicity need to offer the resources children need so they can integrate, if they choose, their birth heritage and their adoptive culture (Waterman et al. 2018). Additionally, parents must acknowledge and prepare children who look physically different to face discrimination (Baden 2016; Cloonan 2022; Waterman et al. 2018; Park-Taylor and Wing 2019) based on their adoption status or physical appearance.

**Author Contributions:** Conceptualization, V.C., T.H. and S.B.; methodology, L.D.; formal analysis, V.C.; data curation, V.C. and T.H.; writing—original draft preparation, V.C. and S.B.; writing—review and editing, V.C., T.H., S.B. and L.D. All authors have read and agreed to the published version of the manuscript.

**Funding:** This research received no external funding.

**Institutional Review Board Statement:** The Institutional Review Board approved the study at the University of the Cumberlands #851-1221 on 10 December 2021.

**Informed Consent Statement:** Informed consent was obtained from all subjects involved in the study.

**Data Availability Statement:** Data sharing is unavailable due to restrictions from the University of the Cumberlands Institutional Review Board.

**Conflicts of Interest:** The authors declare no conflict of interest.

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
