# Peer review of "The Racial and Ethnic Identity Development Process for Adult Colombian Adoptees"

_genealogy, doi:10.3390/genealogy7020035_

Round 1

Reviewer 1 Report

This article explores ethnic/racial identity formations in Colombian adoptees to the USA, through an analysis of narratives of six adoptee interviewees. 

The data and analysis is rich and interesting, and the article builds on existing scholarship and fits in well with contemporary critical adoption studies. The authors' understanding of the prevalence and relevance of unethical and illegal adoptions is appreciated. However, I was curious as to whether they found that their interviewees' discovery of fraudulence in the adoptions impacted their racial/ethnic self identification? The authors' mention this in relation to previous research, and I wondered whether the authors could relate this to their own findings ("Baden et al. (2012) highlight how transra- cial adoptees face challenges regarding their identity development due to the fabricated or absent information on their birth family") 

It also may be helpful to readers unfamiliar with the adoption industry to expand a bit on the nature of unethical adoptions, and adoption corruption in general.

Similarly, I would suggest a few more lines on the history and present situation of Colombia - USA adoptions. 

Particularly for readers outside the context of North America, it may be useful to briefly clarify the meanings of "racial and ethnic" identity formation - as the way race and ethnicity is understood can vary greatly. 

Regarding the Conclusion: Would it be possible to relate the findings to other international adoptees (i.e. not Colombian)? Do the authors have any recommendations for adoptees, adoption rights activists, practitioners or family of adoptees regarding the complexities of adoptee racial/ethnic identity formation? 

In all I found this an interesting article that adds to critical adoption scholarship. I recommend it is published with the minor revisions suggested above.

The article is well-written. The authors might want to edit the introduction section, but apart from that it is fine. 

Author Response

Thank you,

The authors. 

Reviewer 2 Report

Early in the introduction it should be made clear that the adult Colombian Adoptees were adopted to the US as it does not otherwise make sense to immediately take the reader to US context content.

Page 1 lines 23-27 makes reference to the lack of adoption regulation but there is no indication of dates for this or mention of more recent regulation such as that imposed by the Hague Convention on Intercountry Adoption which the US have ratified.

On page 1 line 40 some Colombian adoptions have been described as 'ethical and legal' but there is no clarification as to what constitutes either for the reader to be informed what these are and so understand adoptions that fall short of these terms.

Page 1 lines 41-43 ICAs are casually linked to trafficking without discussion or references to the extensive literature out there on this issue. Suggest you expand this a little or remove, but one unreferenced comment here is neither helpful to an advancement on your theme.

On page 1 line 46, you suggest that the challenges faced by Colombian adoptees are 'unique' . In what ways are they different from those of other Intercountry adoptees? The identity issues faced by your participants and discussed in your article seem similar to those of other  Intercountry and transracial adoptees, so you need to clarify this point further as to what is unique for these particular adoptees.

Page 2 line 67 refers to a 'racial and ethnic identify process' please clarify what you mean here.

On page 2 lines 67-72 details of the researchers are highlighted but then not referred to again in any of the analysis or discussions. If these researcher positionalities are not reflexively used I suggest that this content is moved to the introduction as context setting rather than leaving it in the methods/methodology section.

No ethical discussion is apparent in the article beyond research approval through a review board which - given the sensitivity of the topic for some , some ethics discussion is expected here.

Page 2 line 77 you refer to 'research identities' please clarify what you mean here.

In table 1 on page 3 lines 107-108 you highlight 'Adoptive parents social class' for those outside of the US this needs clarifying as social class is defined in alternative ways in various countries, so this means very little to the non-US reader.

In section 3 on page 4 line 149 you suggest poor post adoption support is available. Is this specific to US? if so be clear that this is the case because some countries have clear structures of post adoption support for adoptees up to the age of twenty-one (and longer for special needs) in some cases (such as UK). So be clear that this lack of support is evident in the US but not necessarily universal.

Page 5 lines 180 'Mestizo' needs clarifying for those outside the US. Page 6 line 239 'la camiseta' also needs explaining.

Claims such as that made on p10 lines 424-425 should perhaps again be situated in US as the literature in Europe is more evident in this regard.

The gender claim made in lines 428 about gender differences is weak given the tiny number of participants and is ill supported by the chosen  and very limited references, given that the Rushton et al study only had women as participants. This claim adds little to your article and is speculative in your suppositions and so would benefit from further study and analysis rather than included here.

Overall, I feel that the small sub-themes in this article are too numerous to really enable the reader to be informed by and have their thinking advanced by the debate and data presented. Each theme is under-explored and insufficiently linked to the wider literature on the topic.

A stronger approach might be to reduce the number of themes in your article and explore them in more detail to be able to highlight the 'Colombian adoptee' experience more. This would serve to separate your research themes from those of previous and existing research which extensively addresses the identity challenges faced by intercountry and transracial adoptees.

The conclusion seems to be a repeat of claims and data detail rather than a robust conclusion to the article.

Author Response

Hello, 

Please see the attachment. Thank you for taking the time to review our manuscript. 

The authors.

Round 2

Reviewer 2 Report

Thank you for sharing your revised paper. I think the changes have improved the clarity and coherence overall and your additional content really helps to contextualise and situate your claims. 

I note your point that reducing the sub-themes would not be true to your participants and I certainly did not intend this with my suggestion. Only that addressing less sub-themes in an individual article allows you more space for depth in your analysis and discussion and can actually increase the number of articles produced from your research in that you are not trying to cover all the findings of your project in one article. However, this is your choice to make.